# Convolutional vs. Recurrent Neural Networks for Audio Source Separation

**Shariq Mobin\*, Brian Cheung\*, and Bruno Olshausen**
Redwood Center for Theoretical Neuroscience
University of California, Berkeley
{shariqmobin,bcheung,baolshausen}@berkeley.edu

## Abstract

We propose a convolutional neural network as an alternative to recurrent neural networks for separating out individual speakers in a sound mixture. Our results achieve state-of-the-art results with an order of magnitude fewer parameters. We also characterize the robustness of both models to generalize to three different testing conditions including a novel dataset. We create a new dataset *RealTalkLibri* which evaluates how well source separation models generalize to real world mixtures. Our results indicate the acoustics of the environment have significant impact on the performance of all neural network models, with the convolutional model showing superior ability to generalize to new environments.

## 1 Introduction

Inferring the individual speaker waveforms that make up a mixture requires strong prior knowledge on the representation of speaker waveforms. Recently, neural network models have been shown to accomplish state-of-the-art results on this task (Hershey et al., 2016; Chen et al., 2017; Isik et al., 2016), see Figure 4.

These models have focused on using Bi-directional Long short-term memory (BLSTM) (Graves et al., 2005) architectures on simulated mixtures of two speakers. However, convolutional neural networks have been shown to exceed the performance of many other neural network architectures (Yu & Koltun, 2015; Sigtia et al., 2016). We show our CNN architecture exceeds the performance of previous models on simulated mixtures and real-world mixtures which we introduce in this work.

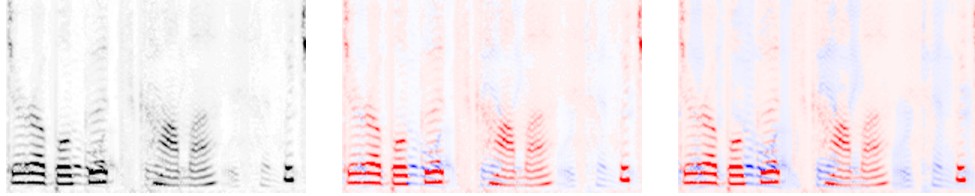

Figure 1: (Left to Right) Spectrogram of the mixture, source estimates using the oracle (red and blue), source estimates using our method.

## 2 Deep Attractor Framework

### 2.1 Embedding the mixed waveform

Chen et al. (2017) propose a framework for single-channel speech separation. $x \in \mathbf{R}^\tau$ is a raw input signal of length $\tau$ and $X \in \mathbf{R}^{F \times T}$ is its spectrogram computed using the Short-time Fourier transform (STFT). Each time-frequency bin in the spectrogram is embedded into a K-dimensional latent space $V = f(X; \theta) \in \mathbf{R}^{F \times T \times K}$ by a transformation $f(\cdot; \theta)$ with parameters $\theta$.

---

\*Equal Contribution

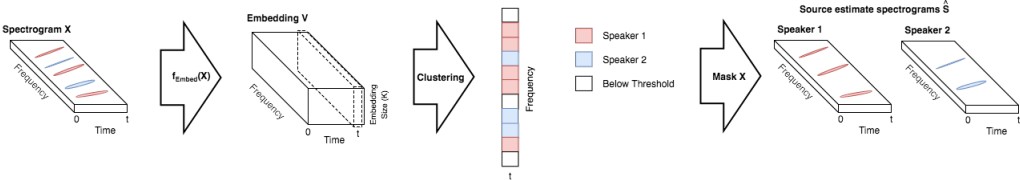

Figure 2: Overview of the source separation process.

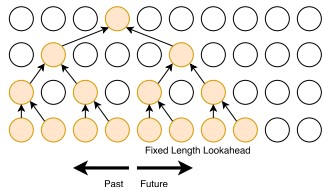

Figure 3: One-dimensional, fixed-lag dilated convolutions used by our model.

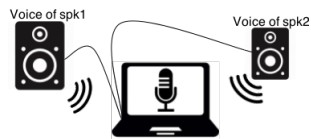

Figure 4: Recording setup for *RealTalkLibri* data. Two computer speakers imitate two human speakers speaking at the same time to a recording device on the computer.

## 2.2 CLUSTERING THE EMBEDDING

We assume that each time-frequency bin can be assigned to one of the $C$ possible speakers. The Ideal Binary Mask (IBM), $Y \in \{0, 1\}^{F \times T \times C}$, is a one-hot representation of this classification for each time-frequency bin. The time-frequency bin is assigned to the speaker who's source target spectrogram, $S \in \mathbf{R}^{F \times T \times C}$, has the highest power. We estimate $Y$ by a mask $M \in (0, 1)^{F \times T \times C}$ computed from the spectrogram $X$.

An attractor point, $A_c \in \mathbf{R}^K$, can be thought of as a cluster center for a corresponding source $c$. Each attractor $A_c$ is the mean of all the embeddings which belong to speaker $c$. In the absence of the oracle mask at test time, the attractor points are calculated using K-means. The mask is computed by taking the inner product of all embeddings with all attractors and applying a softmax, this is then used to calculate a reconstruction of the source target spectrogram:

$$A_{c,k} = \frac{\sum_{f,t} V_{f,t,k} Y_{f,t,c}}{\sum_{f,t} Y_{f,t,c}}; \qquad M_{f,t,c} = Softmax\Big(\sum_k A_{c,k} \times V_{f,t,k}\Big); \qquad \hat{S}_{\cdot,\cdot,c} = M_{\cdot,\cdot,c} \odot X$$

The loss function $\mathcal{L}$ is the mean-squared-error (MSE) of the source estimate spectrogram, $\hat{S}$ and the supervised source target spectrogram, $S$. See Figure 2 for an overview of our source separation process.

## 2.3 OUR MODEL

Our network consists of 13 dilated convolutional layers (Yu & Koltun, 2015) made up of two stacks, each stack having its dilation factor double each layer. Our filters were of shape (3, 3). Batch Normalization (Ioffe & Szegedy, 2015) was applied to each layer and residual connections (He et al., 2016) were used at every other layer. Our model has a fixed-lag response of 127 timepoints ($\sim$ 1s, see Figure 3).

## 3 EXPERIMENTS

First, we construct artificially mixture data sets according to the standard procedure introduced in Hershey et al. (2016). Additionally, Figure 4 shows how we generated our more realistic *RealTalkLibri* mixture data. Specific details can be found in Appendix A.

We begin by evaluating the models on the WSJ0 test dataset as in Chen et al. (2017). Our state-of-the-art results are shown in the WSJ0 section of Figure 6. Our model achieves the best score using a

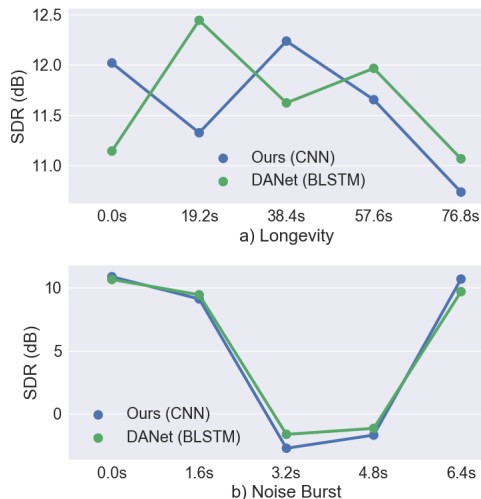

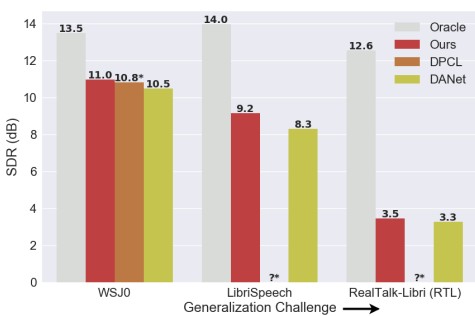

Figure 6: Results of models tested on WSJ0 simulated mixtures, LibriSpeech simulated mixtures, and our RealTalkLibri (RTL) dataset. *: Model has a second neural network to enhance the source estimate spectrogram and is therefore at an advantage. Model wasn't available online for testing against LibriSpeech or RTL.

Figure 5: Results of the length and noise generalization experiments. For both plots, we plot the SDR starting at the time specified by the x-axis up until 400 time points ($\sim$ 3s) later.

factor of ten fewer parameters than DANet (1.6M vs. 16M). We test the generalization abilities of the models by:

Length Generalization: how well these models work under time-sequences 25x longer than they are trained on, i.e. $T = 10000$ ($\sim$ 80s). Results are shown in Figure 5a.

Noise Generalization: how the models respond to small bursts of input data far outside of the training distribution. We took sequences of length $T = 1200$ ($\sim$ 9s) and inserted white noise for 0.25s in the middle of the mixture to disrupt the models process. Our results are shown in Figure 5b.

Data Generalization: how well the models generalize to data progressively farther from their training distribution. We trained all the models on the WSJ0 training set and then tested on the WSJ0 test set, the LibriSpeech test set, and *RealTalkLibri* test set. Our results are shown in Figure 6.

## 4 DISCUSSION

Here we explored using convolutional neural networks as an alternative model for source separation. Our state-of-the-art results on the WSJ0 dataset using a factor of ten fewer parameters show that convolutional models are both more accurate and efficient for audio source separation.

In order to study the robustness of all models we tested their performance under three different conditions: longer time sequences, intermittent noise, and datasets not seen during training. While the BLSTM generalized suprisingly well in the first two conditions, our convolutional neural network also generalized better to both the LibriSpeech dataset and the *RealTalkLibri* dataset we introduced here. Models which are robust to new datasets as well as the deformations caused by the acoustics of different environments will be critical to progress in audio source separation. Our *RealTalkLibri* dataset complements other real-world speech datasets Barker et al. (2015); Kinoshita et al. (2013) by additionally providing approximate ground truth waveforms for the mixture which was not available before.

Looking forward, we aim to improve the generalization ability on examples such as those shown in Figure 7 by introducing a training set for *RealTalkLibri*, developing more robust model architectures, introducing regularizers for the structure of speech, and creating powerful data augmentation tools. We also believe models which can operate under an unknown number of sources is of utmost importance to the field of audio source separation.

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

## A    *RealTalkLibri* DATASET

The main motivation for creating this dataset is to record mixtures of speech where the acoustics of the room deform a high quality recording into a more realistic one. While datasets of real mixtures exist, there exists no dataset where the ground truth source waveforms are available, only the transcription of the speakers words are given as target outputs Kinoshita et al. (2013); Barker et al. (2015). In order to understand how well our model generalizes to real world mixtures we created a small test dataset for which there is ground truth of the source waveforms.

The *RealTalkLibri* (RTL) test dataset was created starting from the test-clean directory of the open LibriSpeech dataset Panayotov et al. (2015) which contains 40 speakers. We first downsampled all waveforms to 8kHz. Each mixture in the dataset was created by sampling two random speakers from the test-clean partition of LibriSpeech, picking a random waveform and start time for each, and playing the waveforms through two Logitech computer speakers for 12 seconds. The waveforms of the two speakers were played in separate channels linked to a left and right computer speaker, separated from the microphone of the computer by different distances. The recordings were made with a sample rate of 8kHz using a 2013 MacBook Pro (Figure 4). In order to obtain ground truth of the individual speaker waveforms each of the waveforms was played twice, once in isolation and once simultaneously with the other speaker. The first recording represents the ground truth and the second one is for the mixture. To verify the quality of the ground truth recordings, we constructed an ideal binary mask $Y$ which performs about as well on the previous simulated datasets, see the Oracle performance in Figure 6. The *RealTalkLibri* data set is made up of two recording sessions which each yielded 4.5 hours of data, giving us a total of 9 hours of test data. Our code for making this data set is available at https://github.com/not/accepted/yet and the data generated is available at https://www.not.accepted.yet.

### A.1    MISTAKES ON *RealTalkLibri* DATASET

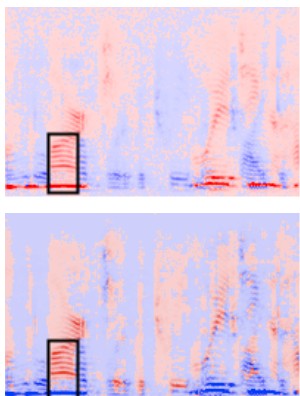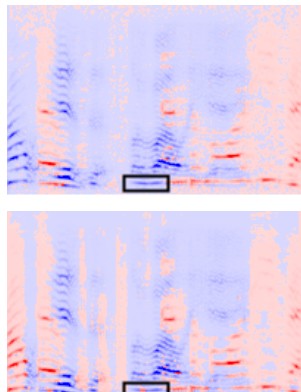

Figure 7: Two examples of source separation on the *RealTalkLibri* dataset (left and right column). For each example we plot the source estimates using the oracle (row 1), and the source estimates using our method (row 2). Notice that the model still has difficulty maintaining continuity of speaker identity across frequencies of a harmonic stack (left column) and across time (right column).

In Figure 7 we visualize the mistakes our network makes under the *RealTalkLibri* dataset. The first example indicates that the model does not have a strong enough bias to the harmonic structure contained in speech, it classifies the frequencies of the fundamental to a different speaker than the harmonic frequencies of that fundamental. The second example indicates that the model also has issues with temporal continuity, the speaker identity of fixed frequency bins varies sporadically. This indicates that there is still room to improve generalization in these models by modifying model architecture and adding regularization.

