# OpenReview forum: "Convolutional vs. Recurrent Neural Networks for Audio Source Separation"
_ICLR.cc/2018/Workshop — Reject_

### Official Review · AnonReviewer1 · 2018-03-08
**comments**

**Rating:** 6
**Confidence:** 4

**Review:**

The paper investigates using dilated CNNs for speech separation and compares with BLSTM on various tasks. The technique itself is not novel but the application on speaker modeling in the separation task is sort of interesting.   The experimental results are supportive.  One concern however is about the computational complexity of dilated CNNs.  The authors should say a few words about it.  Furthermore, there are a few acronyms used without definitions. (e.g. SDR, DPCL, etc. ). Last but not least, I wonder if the authors can talk a bit more on their RealTalkLibri dataset such as how the data is transcribed, the environments (SNR, noise types) and speakers (gender, age, dialect, etc.)

---

### Official Review · AnonReviewer2 · 2018-03-09
**Reasonable modification to the DANet, new dataset with a few issues**

**Rating:** 6
**Confidence:** 5

**Review:**

This paper describes a modification of the deep attractor network (DANet) source separation framework to use a deep CNN instead of a deep BLSTM.  It also includes the recording of a 4.5 hour dataset of speech played through computer speakers in a real room. The new network appears to achieve similar or slightly better performance to the baseline DANet and is reported to use 1.6M parameters vs DANet's 16M.

There are a few issues with the paper.  The experiments show a "DPCL" system in certain conditions, but this is never defined.  It is presumably the original deep clustering system.  In the experiments shown in figure 4, it is not clear what the x-axis refers to.  For the noise experiment, is that the position of the noise in a single utterance?  Is it the duration of the noise?  Is it averaged over several utterances?  How many utterances were involved in the duration experiment?  The duration of the signals does not affect the CNN, so it is only for the purposes of the clustering, right?  This should be stated.

The abstract claims that "Our results indicate the acoustics of the environment have significant impact on the performance of all neural network models" this is a well-known finding, c.f., the REVERB challenge results, but the current paper's results do not show that this is the case for ALL neural networks, just these two.  It also states "the convolutional model showing superior ability to generalize to new environments" but from the results on the new RealTalkLibri dataset this is 3.5dB SDR vs 3.3dB SDR, which is rather small.

In addition, the recording conditions of the new data should be described in more detail.  What kind of room was this?  How big was it?  What was in it?  What was its reverberation time (RT60)?  Where were the speakers placed?  Were they moved between utterances or at all?  Was there ambient noise in the room?  If so, was it stationary or non-stationary?

The playback of sounds over speakers is not really "real" per se, but it is perhaps slightly more real than recording impulse responses and simulating reverberation, and it is more real than just adding two dry recordings together.  It is less real than recording actual people talking because actual people move when they are talking, but this brings its own problems, as noted in the paper, of not having ground truth.

The use of a separate playback of the same sound as "ground truth" for the mixture seems a bit questionable.  Why not just record each sound separately through the channel and then add them after the fact?  Sounds do add in air and in microphones linearly, unless there is clipping.  You can make multiple recordings of each sound if you want, or just add them in many combinations.

Overall, the model is a reasonable modification to the DANet, although if its main feature is having fewer parameters, then the experiments could explore that further by measuring the effect of different numbers of parameters in each model.  And the dataset is promising, but could be strengthened slightly in its construction and more so in its description.

---

### Official Review · AnonReviewer3 · 2018-03-10
**Existing CNN architecture (dilated conv. nets) paired with existing source separation framework (attractor nets) and performs on-par to slightly better than RNN version. Lower novelty. Marginal Reject.**

**Rating:** 5
**Confidence:** 4

**Review:**

The authors utilize convolutional rather than recurrent networks in the recently introduced
attractor network framework for source separation, and show that it performs slightly better, despite
using 10x less parameters. Furthermore, their cnn architecture generalizes somewhat better to mismatched test data, and source separation dataset based on librispeech is defined and released for use.

While the results in this paper will be of great interest to audio source separation researchers, wrt research on learning feature representations the paper is somewhat lacking in novelty, since they are simply pairing an existing convolutional architecture (Fisher et al., 2015) with a recently introduced but existing source separation framework (Chen et al, 2017).

---

### Decision · Program_Chairs · 2018-03-20
**ICLR 2018 Workshop Acceptance Decision**

**Decision:**

Reject

**Comment:**

Based on the reviews, this paper has not been accepted for presentation at the ICLR workshop. However, the conversation and updates can continue to appear here on OpenReview.